# Preventing Stress among High School Students in Denmark through the Multicomponent Healthy High School Intervention—The Effectiveness at First Follow-Up

**DOI:** 10.3390/ijerph20031754

**Published:** 2023-01-18

**Authors:** Camilla Thørring Bonnesen, Lau Caspar Thygesen, Naja Hulvej Rod, Mette Toftager, Katrine Rich Madsen, Marie Pil Jensen, Johanne Aviaja Rosing, Stine Kjær Wehner, Pernille Due, Rikke Fredenslund Krølner

**Affiliations:** 1National Institute of Public Health, University of Southern Denmark, 1455 Copenhagen, Denmark; 2Section of Epidemiology, Department of Public Health, University of Copenhagen, 1014 Copenhagen, Denmark; 3Department of Sports Science and Clinical Biomechanics, University of Southern Denmark, 5230 Odense, Denmark

**Keywords:** school-based intervention, high school, stress, effectiveness

## Abstract

Stress is a widespread phenomenon and young people especially are experiencing high levels of stress. School-related factors are the most frequently self-reported stressors among adolescents, but few interventions have targeted the school environment. This study evaluated the effectiveness of the Healthy High School (HHS) intervention on stress at a 9-month follow-up. The study included 5201 first-year high school students (~16 years) in Denmark. Participating schools were randomized into the HHS intervention (N = 15) or control group (N = 15). Baseline measurements were conducted in August 2016 and the follow-up was conducted in May 2017. The intervention was designed to promote well-being (primary outcome) by focusing on physical activity, meals, sleep, sense of security, and stress (secondary outcomes). The intervention comprised: structural initiatives at the school level; a teaching material; peer-led innovation workshops; and a smartphone app. The 10-item Perceived Stress Scale was used to measure stress. Intervention effects on perceived stress were estimated using an intention-to-treat approach with multiple imputations of missing data and multilevel general linear regression modelling. A total of 4577 students answered the baseline questionnaire. No statistically significant difference was found in stress between students at intervention and control schools at the follow-up (mean score: 16.7 versus 16.7, adjusted b = 0.42, 95% CI: −0.16;1.00). The HHS Study is one of the first large randomized controlled trials targeting school environmental stressors. Potential implementation failures and the failures of the program theory are discussed.

## 1. Introduction

Stress seems to be a significant worldwide problem and young people especially report high stress levels [1,2]. Moreover, today’s generation of young people are more stressed than those of previous generations [3,4]. In a Danish national youth survey, 55% of high school girls and 35% of high school boys reported to feel stressed daily or weekly [2]. Long-term stress among adolescents has been linked with negative mental health outcomes such as depression and anxiety [5,6,7,8,9,10], suicidal behavior [11,12], disturbed sleep [13,14,15], and a wide range of unhealthy behaviors including physical inactivity, unhealthy eating [16,17], heavy alcohol consumption [15,16,18], and tobacco smoking [16,18,19].

The transition to high school is a challenging period for many adolescents as there is a radical shift in school context [20], a diminished reliance on parents, and wider connections with peers [21]. While many adolescents make smooth transitions to high school, the transition can develop into a stressful and turbulent time for others [20,22]. The school and the psychosocial working environment it provides for the students has long been an area of central importance, and numerous studies have identified school-related factors as important stressors among adolescents [1,23,24].

Schools are considered as important settings for public health initiatives for children and adolescents. They spend a considerable amount of time at school, and the school provides an opportunity to reach almost everybody within the relevant age group [25]. In addition, previous studies have shown that young people may have a low interest in issues of health and stress prevention [26] and rarely seek help when distressed [27], making it difficult to reach them by voluntary initiatives after school. 

Internationally, there has been a growing body of studies investigating the effects of educational strategies to prevent stress in school settings. For example, school-based interventions targeting stress management (e.g., problem solving, social skills training, mindfulness, relaxation techniques, and time management) in children and adolescents seem to be effective in reducing stress symptoms, self-reported stress, and cardiovascular parameters of stress (e.g., blood pressure) and in improving coping, academic, and social skills [28,29,30,31,32,33,34,35]. 

While educational initiatives focus on individual behavior changes, there are many sources of stress that involve the structure or culture of an organization (e.g., workplace or school) which are less amendable at the individual level [36]. Moreover, reviews evaluating the effectiveness of stress preventive interventions in the workplace have concluded that interventions combining environmental (changing organizational practices or altering the physical or psychosocial work environments) and educational strategies have the greatest potential to prevent stress [37,38,39]. Previous research on stress among students have advocated for interventions targeting the school environment [23,40,41,42]. Despite knowledge of the importance of the psychosocial working environment that schools provide for the students, there is a lack of school-based studies combining educational and environmental strategies. We are not aware of any such studies in the high school context. 

The Healthy High School (HHS) Study was developed to promote well-being (the primary outcome) among first-year high school students (16-year-olds) in Denmark using a combination of educational and environmental initiatives. A thorough needs assessment (literature reviews and quantitative and qualitative research) among the target population identified stress prevention and the promotion of regular meals, physical activity (PA), sleep, and students’ sense of community at school as five important pathways to achieve a higher level of well-being. The purpose of this study was to examine the effects of the HHS Study on students’ perceived stress at the 9-month follow-up. The primary outcome of the trial (well-being) as well as the outcomes of the four other pathways to well-being will be reported elsewhere.

## 2. Materials and Methods

### 2.1. Study Design and Participants

The HHS Study was a school-based cluster-randomized controlled trial involving first-year high school students (age ≈ 16) in Denmark. The trial is registered in Current Controlled Trials (ID: ISRCTN43284296, 28 April 2017) and has been described in detail elsewhere [43]. We invited high schools that had previously participated in the Danish National Youth Study 2014 (DNYS), a national web-based survey among 70,674 high school students (average age of 17.9 years). The DNYS questionnaire included 250 core questions concerning health behavior, health, and well-being [44]. Thirty-one of 92 eligible high schools agreed to participate (33.7%) in the HHS Study and were randomized into the intervention group (*n* = 16) or control group (*n* = 15) by computer-based random number generation. After randomization, one high school from the intervention group withdrew from the study, leaving 30 high schools in the final trial (Figure 1). A detailed description of the flow of high schools and students and the eligibility criteria are provided in the study protocol [43].

### 2.2. The HHS Intervention

The overall aim of the HHS Study was to promote well-being (the primary outcome) among first-year high school students in Denmark. We used the Intervention Mapping protocol [45] to plan the intervention, implementation, and evaluation of the study in a systematic fashion based on theory, evidence, and best practice from Danish high schools [43]. As part of the Intervention Mapping protocol, we conducted a comprehensive needs assessment from February 2014 to October 2015 including (1) 16 focus group discussions with students (*n* = 74), 2 focus group interviews (*n* = 7), 5 individual interviews with teachers, student counsellors, and principals at high schools, and 2 telephone interviews with canteen managers, (2) an epidemiological assessment using existing questionnaire data from the DNYS including more than 70,000 high school students, (3) literature reviews of determinants and previous school-based multicomponent or multiple health behavior change interventions, and (4) a brainstorm session with students and school staff to explore the feasibility of early intervention ideas and high schools’ implementation capacity. The needs assessment identified stress prevention and the promotion of regular meals, PA, and sleep as well as students’ sense of community at school as five important pathways (secondary outcomes) to achieve a higher level of well-being. 

The HHS intervention consisted of four main intervention components which combined environmental and educational intervention strategies and were designed to change the important and modifiable determinants of the five secondary outcomes: (1) structural policies and initiatives at the school level, (2) a teaching material, (3) a peer-led innovation workshop and derived activities, and (4) a smartphone app. The intervention was implemented among first-year high school students in the school year 2016/17 and lasted for nine months (August 2016–May 2017). The intervention components are described below and presented in Table 1.

#### 2.2.1. Structural Policies and Initiatives for a Supportive School Environment 

A total of 16 initiatives (9 mandatory and 7 optional) aimed to create a healthy and supportive school environment. Four initiatives aimed directly at preventing stress including the development of a stress policy, introduction of annual coursework plans for each school class, half-yearly counselling sessions for all students, and a two-hour time management session. The initiatives were described in an online catalogue (password-protected) addressed to student councils, school managements, teachers, student counsellors, canteen staff, and school caretakers (accessible in Danish here: www.sdu.dk/da/sif/bgym, accessed on 17 January 2023). Danish high schools have different physical, economical, and organizational opportunities for changing the school environment, thus the catalogue comprised different suggestions on how to implement the initiatives. 

#### 2.2.2. Teaching Material 

In close collaboration with high school teachers, the HHS teaching material (accessible in Danish here: www.sdu.dk/da/sif/bgym, accessed on 17 January 2023) was designed to change social norms and cognitive factors such as knowledge, awareness, and outcome expectancies through educational activities in four different subjects (Danish, Social Studies, Physical Education and Sport, and Introduction to Natural Science) which were common for all first-year students. It comprised a total of 17 lessons (1440 min) and optional lessons within a one-week Multi-Subject Coursework (a Multi-Subject Coursework combines curricular activities within two subjects and focusses on general study preparation and academic methods). All lessons were designed to cover the official learning goals defined by the Danish Ministry of Education and were intended to replace some of the standard lessons for the included subjects. Stress was addressed in five lessons (405 min) and included material on the physiology of stress, general stress symptoms, short- and long-term health consequences of stress, external and internal stressors, and the interrelations between stress, meal habits, sleep, and PA. Teachers were introduced briefly to the material at a pre-intervention kick-off conference but were expected to be able to deliver the activities without prior training.

#### 2.2.3. Peer-Led Innovation Workshop and Derived Activities

The peer-led innovation workshop aimed at inspiring high school students to develop and implement new activities to promote PA and a sense of community at school and did not address stress prevention directly. The workshop was facilitated during school hours by university students in Sport Science and Health (peer mentors) at all intervention schools within the first months of the intervention. The workshop included two parts of 90 min each, held at two separate days with one or two weeks in between. The workshop was based on a peer-led approach, student co-determination, innovation techniques, and local solutions. Teachers were encouraged to support the students to refine their ideas and to initiate the new activities at school. The high school students had the opportunity to apply for a grant to establish their activities (up to 40,000 DKK per high school). 

#### 2.2.4. Smartphone App

The aim of the HHS smartphone app was to promote well-being and support students’ behavior changes outside school hours by influencing their knowledge and awareness of stress, sleep, PA, and meal habits. The app included articles, tracking options, recipes, training programs, quizzes, time management techniques, mindfulness practices, and an option to sign up for four different eight-week courses on how to prevent stress, increase PA, and improve sleep or meal habits. The app was developed in collaboration with a company which specialized in designing personalized digital health programs. The research group developed and wrote the main part of the content, while the app company designed the layout and technical functions of the app. The content about stress was based on academic books, scientific literature, and recommendations from the Danish Health Authority. The focus was on writing short texts in a non-academic language, inspired by youth magazines to appeal to young people. Contrary to the original implementation plan, the app was introduced to the students 12 weeks after the introduction of the intervention due to a delayed delivery from the app developing company. Students were introduced to the app by teachers at the school or by a member of the research group. Students received written information about the features of the app and were shown a tutorial video to illustrate how the app worked. We used gamification to motivate participation and posted three competitions on the app from November 2016 to April 2017 to encourage more students to download and use the app.

### 2.3. Control Group

The control schools received no intervention and were encouraged to continue as usual. A refined version of the intervention material was offered to the control schools in the summer of 2020.

### 2.4. Data Collection

The baseline survey was conducted at the beginning of the school year (August 2016) and included 5201 students, of which 4577 responded to the questionnaire (response rate = 88.0%). The first follow-up was conducted at the end of the school year (May 2017) and included 4512 students, of which 3426 responded to the questionnaire (response rate = 75.9%) (Figure 1). A week before the data collection, information materials aimed at each school class were sent to school coordinators. Students answered the questionnaire in class after a standardized instruction given by a teacher. The students were instructed to fill out the questionnaire individually without talking to classmates within one school lesson of 45 min. Teachers were asked to be present during the questionnaire session and remind absent students to answer the questionnaire at a later point in time.

### 2.5. Outcome Measure

Perceived stress was measured by the Perceived Stress Scale 10-item version (PSS-10) [46]. The PSS-10 is a self-reported measure of global stress developed to assess the extent to which people find their life unpredictable, uncontrollable, and overloaded. The scale consists of ten items rated on a 5-point Likert scale ranging from “never” to “very often”. The responses of the positively stated items were reversed, and all item scores were added together to sum a score with a potential range from 0 to 40, with a low score being favorable. 

### 2.6. Covariates 

Parental OSC was measured by the students’ response to the questions: ‘Does your father/mother have a job?’ and ‘What is his/her job title?’. The research group coded the responses in accordance with the Danish Occupational Social Class Measurement into I (high) to V (low) [47]. We added a category OSC VI for parents outside the labor market receiving unemployment benefits, a disability pension, or other kinds of transfer incomes. Each student was categorized by the highest-ranking parent into high OSC (I–II, e.g., professionals and managerial positioners), middle OSC (III–IV, e.g., technical and administrative staff, and skilled workers), low (V, unskilled workers and VI, outside labor market), and unclassifiable. Several studies have demonstrated that school children from the age of 11 can report their parents’ occupation with a fair validity [48,49,50,51,52,53] and that parental OSC is an appropriate socioeconomic position indicator in studies of adolescents [54]. Immigrant background was based on three items that assessed the country of birth of the student and their mother and father. Based on the definitions of Statistics Denmark [55], each student was categorized as being Danish (having at least one parent born in Denmark regardless of own country of birth), a descendant (born in Denmark by parents born outside of Denmark), or an immigrant (born abroad by parents born outside of Denmark). Major life events were measured by asking the students whether they had experienced serious personal injury or illness, serious injury or illness of a close family member, death of a close family member, death of a close friend, divorce of parents, parents having serious economic problems, bullying, loss of a close friendship, and break up with girlfriend/boyfriend with response options: “No”, “Within the past year”, and “More than a year ago”. A sum score ranging from 0 to 9 was created, with a low score being favorable.

### 2.7. Statistical Analyses 

A total of 904 students had no baseline data either because they did not complete the baseline questionnaire (N = 624) or were invited for follow-up only (N = 280). They were excluded from this study. 

Descriptive baseline data were used to (1) characterize the study population, (2) compare high schools and students from the intervention and control groups, and (3) explore if the characteristics of students lost to follow-up differed between the randomized groups (attrition bias). To account for the nested design, multi-level generalized linear models were used to estimate the association between the intervention and perceived stress. The analysis was performed as a three-level model with students (level 1) nested within classes (level 2) and classes nested within high schools (level 3), allowing for a correlation between students from the same class and high school. The model included the independent variable and added baseline measures of PSS-10, and three covariates highly associated with the outcome: gender, OSC, and major life events. The covariates were chosen a priori and were included to increase the precision of the effect estimates [56]. Class and school were included as the random effects. Intervention group and covariates were included as the fixed effects. We used an intention-to-treat approach with multiple imputation of missing data at the follow-up under the missing at random assumption. The imputation was based on variables from the baseline questionnaire which were expected to be correlated with the six pre-planned outcome measures, lost to follow-up, or both, e.g., gender, socio-economic position, and other health behaviors. We performed two rounds of 20 imputations, as suggested by Graham (2012) [57], to account for the multi-level data structure.

The statistical analyses were carried out using SAS, version 9.4 (SAS Institute Inc, Copenhagen, Denmark). A significance level of 0.05 was chosen a prior. The analyses were pre-specified in a statistical analysis plan, and all co-authors agreed to the plan before the analyses were carried out. 

#### Sensitivity and Exploratory Analyses 

A sensitivity analysis was carried out on complete cases without imputation of missing values. Universal interventions are designed to target all individuals in a population and are not directed at a specific group at risk. However, previous studies have found that universal interventions are effective only for a subsample of the population and thus may widen inequalities (inverse equity hypothesis) [58,59,60]. Therefore, we performed exploratory stratified analyses to investigate whether the intervention was equally effective for boys and girls and students of low and high OSC.

### 2.8. Ethical Considerations 

The HHS Study adheres to all Danish ethical standards. The Regional Scientific Ethical Committee, the Capital Region of Denmark, reviewed the HHS Study and concluded that formal ethical approval was not required (J.nr. 16018722). The trial is registered and listed in the Danish Data Protection Agency (J.nr. 2015-57-0008). Written information was sent to principals, teachers, and student councils at all high schools which were invited explaining the implications of participating in the study. The participants were informed that their participation was voluntary, that their information would be used for research purposes only, and that their data would be treated confidentially. By participating, they gave consent that their data could be used for research.

## 3. Results

### 3.1. Baseline Characteristics of Students

There were no socio-demographic or outcome differences between the intervention and control groups at the baseline (Table 2). The mean age was 16.2 years and most students were females (63.0%) and of Danish origin (85.3%). Around half of the students were categorized with high OSC (47.3%). The mean PSS-10 score was 14.0, and half of the students reported moderate or high levels of stress (data now shown). 

### 3.2. Attrition Analysis

Out of the 4577 included students, 1551 students (33.9%) were lost to follow-up either because they moved to another school/dropped out of high school (*n* = 689) or had missing data on one or more of the items used in the complete case analyses (862). The student attrition was similar in intervention and control groups according to the baseline gender, age, OSC, and PSS-10. Non-participating students from the intervention group were more likely to be descendants compared to the control group. Students from the control group had on average experienced more major life events within the past year compared to the intervention group (Table 2).

### 3.3. Effect of the HHS Intervention on Perceived Stress

The mean PSS-10 score increased from 13.8 (SD = 6.5) at the baseline to 16.7 (SD = 6.8) at the follow-up among students at the intervention schools, and from 14.2 (SD = 6.4) to 16.7 (SD = 6.7) among students at the control schools. 

In the primary intention-to-treat analysis with an adjustment for the baseline covariates, we found no significant between group difference on perceived stress at the follow-up [b = 0.42, 95% CI: −0.16;1.00]. Similarly, the sensitivity analyses of complete cases did not show any evidence of an effect on perceived stress either [b = 0.59, 95% CI: −0.15;1.33] (Table 3).

### 3.4. Explorative Subgroup Analyses

The explorative sub-group analyses showed that the mean PSS-10 score increased from the baseline to the follow-up among students in all sub-groups. The explorative subgroup analyses showed no effect of the HHS intervention on perceived stress across gender and OSC (Table 4).

## 4. Discussion

The HHS Study is one of the first studies aimed at preventing stress among young people in high school combining educational and environmental intervention strategies. We found no effect of the HHS intervention on perceived stress in the total study sample nor in specific subgroups. The systematic reviews of school-based interventions for stress prevention concluded that educational strategies show some positive effects in the short term [28,30,31,32,33,34,35]. These studies were limited by small sample sizes, the diversity of the study samples, and a variety in the intervention design and outcome measures. One recent review suggests that school-based interventions targeting psychological stress are not effective in community samples and that only selected students benefit from such interventions [32]. Contrary to these studies, the HHS Study recruited a large number of high schools and students and evaluated the effect of the intervention in the strongest study design, minimizing the risk of self-selection bias to the intervention and alternative causes of the effect. 

The mean PSS-10 score increased from the baseline to the follow-up among students at the intervention and control schools. This was expected as the baseline measurement was conducted in the beginning of the school year, immediately after the summer holidays, while the follow-up was conducted close to the beginning of the revision period preceding exams, which is a stressful time for many students. The aim of the intervention was, therefore, to achieve a lower increase in stress levels among students in the intervention group, but the mean PSS-10 score increased equally in both groups.

### 4.1. Discussion of Mechanisms Underlying the Effect Evaluation Findings

The lack of effect of the HHS intervention on perceived stress may be caused by a weakness in the intervention design (theory failure), a low implementation of the intervention (implementation failure), and/or a combination of both. 

#### Design Issues

Choice of practical intervention strategies: The HHS Study was inspired by previous studies showing that school-based multi-component interventions targeting multiple adolescent behaviors simultaneously seem to be the most effective way to promote well-being and healthy behaviors among adolescents [61]. Furthermore, a comprehensive needs assessment among the target group revealed that student well-being was challenged by a combination of high stress levels, poor sleep, and infrequent meals and PA. Likewise, peer relations played an important role for student well-being and school satisfaction [43]. Grounded in the multiple behavior focus and socio-ecological approach, we expected synergistic effects between the intervention components on the secondary outcomes, as well as synergistic effects between the secondary outcomes on the primary outcome (well-being). Moreover, we did not want to overload the high schools and thus challenging the implementation process. Consequently, each secondary outcome was targeted by a few initiatives, e.g., only 5 of 17 lessons in the teaching material were specifically designed to prevent stress, and the time management course was conducted at the beginning of the school year as a one-time event only. It is possible that initiatives need to be more continuous and intensive to prevent stress among students, e.g., good time management skills might need to be reminded and trained continuously. 

The teaching material was developed in close collaboration with high school teachers to ensure that the material appealed to the target group and covered the learning goals for the included subjects as defined by the Danish Ministry of Education. It was, however, more challenging than expected to integrate stress and health in the academic curriculum. The teachers were provided with templates which specified important change objectives of the HHS Study to be achieved during each lesson. Some of the involved teachers did not find the health-related topics relevant to their subject. Particularly, teachers found it irrelevant to incorporate elements where students were encouraged to reflect on their own behavior (to raise awareness) and set goals for behavior change. Thus, most lessons ended up targeting knowledge only. The effect of a single behavior change technique may be very small and insufficient to produce a behavior change [62]. 

Selection of determinants to be addressed by the intervention: Intervention strategies were developed to address the important and changeable determinants of stress identified in the needs assessment. There have been several education reforms in Denmark during the last decade, increasing the pressure on young people, e.g., (1) stricter entry requirements for higher education programs, (2) a ‘speed start bonus’: if students apply for a higher education program within two years of completing upper secondary education, their grade point average (entrance qualification for higher education programs) will be multiplied by a factor of 1.08, and (3) an A-level bonus: if students take an extra A-level subject during upper secondary education, their grade point average will be multiplied by a factor of 1.03. Furthermore, today’s society is characterized by a culture that values competition, an extreme focus on performance, and a constant search for perfection in every part of life [63,64,65]. These themes were also present when we interviewed students and teachers as a part of the needs assessment and process evaluation [66,67]. Such determinants are closely related to stress but are difficult to change in a school-based intervention. 

### 4.2. Implementation Issues

Compared to the stress-specific interventions described previously [28,29,30,31,32,33,34,35], the HHS intervention might have been too complex for high schools to implement. It was designed to promote well-being by changing not only stress but also multiple health behaviors and a sense of community; it targeted both individual-level factors and the school environment; it comprised multiple intervention components; and it allowed for a high degree of flexibility in its implementation. The process evaluation of the stress preventive initiatives showed that the implementation of all stress preventive initiatives was weak and varying by schools and classes, implying low to moderate doses of both individual- and environmental-level initiatives [68]. The main barriers for implementation were time issues, economic restraints, competing interests, and project fatigue [66]. Moreover, contrary to the original implementation plan, the app was introduced to the students 12 weeks after the introduction of the intervention due to a delayed delivery from the app developing company which hindered the students’ uptake of the app. The lack of effect may thereby reflect an implementation failure [69].

### 4.3. Differential Intervention Effects 

In agreement with a meta-analysis by Tinner et al. (2018) [70], we found no differences in the intervention effects on perceived stress by OSC in our explorative sub-group analyses. The meta-analysis included only four studies as most of the identified studies did not report on socio-economic position. As the HHS Study was designed as a universal, school-based intervention targeting all first-year students at participating high schools, we did not expect differential intervention effects.

### 4.4. Strengths and Limitations

The strengths of this study include (1) a systematic, transparent planning process guided by the Intervention Mapping protocol and an explicit program theory informed by a thorough needs assessment, theoretical behavior change techniques, and the best available evidence; (2) a multi-component intervention of a long duration targeting both school environmental stressors and student knowledge, attitudes, and skills; (3) a large sample size of schools and students; (4) a successfully randomized controlled trial design; (5) a high response rates among students; (6) the use of a validated outcome measure; and (7) the use of statistical methods to take the clustering structure of the data and the attrition into account.

There are some limitations to the study. First, the enrolment rate of high schools was low. Only one third of invited high schools accepted the study invitation. Principals’ main reasons for declining the study invitation was a lack of time due to external ministerial demands such as spending cuts and the implementation of a new comprehensive education reform. In a previous study, we found that students at HHS schools did not differ noticeably from students at non-participating high schools with regard to their gender, well-being, sense of community, PA, sleep, stress, and meal habits [71].

Second, one third of students were lost to follow-up because they no longer attended one of the participating high schools at the time of the follow-up, or because their high schools were unable to find time to conduct the survey among students. Overall, the attrition was similar among intervention and control groups, but non-responders from the intervention group were more likely to be descendants compared to the control group. On the other hand, the students of the control group had experienced more major life events within the past year compared to the intervention group. We handled the attrition by using multiple imputation of missing data. The complete case analysis (sensitivity analysis) showed similar findings, which may reflect that attrition bias was not a major problem [72].

## 5. Conclusions

Previous studies have advocated a need for interventions addressing the school environment to prevent stress among students. As part of the larger multi-component school-based HHS intervention to promote well-being (the primary outcome), educational and environmental strategies were developed to prevent stress among high school students (16-year-olds) in Denmark. In this study, we evaluated the effect of the HHS intervention on students’ perceived stress at a 9-month follow-up. We found no evidence of an effect of the intervention in the total study population nor in specific subgroups. The lack of effect may be due to the low implementation of several of the main stress preventive initiatives and limitations in the design of the intervention program; inducing a change in stress levels may require more intensive interventions. Future interventions targeting students in high schools should carefully consider including a feasibility study to identify challenges with implementing interventions and delivering trials in a high school setting. 

The descriptive data showed that half of students reported moderate or high levels of perceived stress. This highlights the continued need for effective interventions targeting stress among young people in Denmark. Future interventions could benefit from including the whole system of young people’s lives and focus more on several structural levels apart from just the school setting. 

An important next step for the HHS Study will be to evaluate the association between implementation and perceived stress in the intervention group and to examine the school-level characteristics of schools in groups of a high compared to a low implementation. These analyses can contribute further knowledge regarding the impact of the HHS intervention and help clarify whether the lack of effect was caused by an inadequate implementation or the poor quality of the intervention. 

## Figures and Tables

**Figure 1 ijerph-20-01754-f001:**
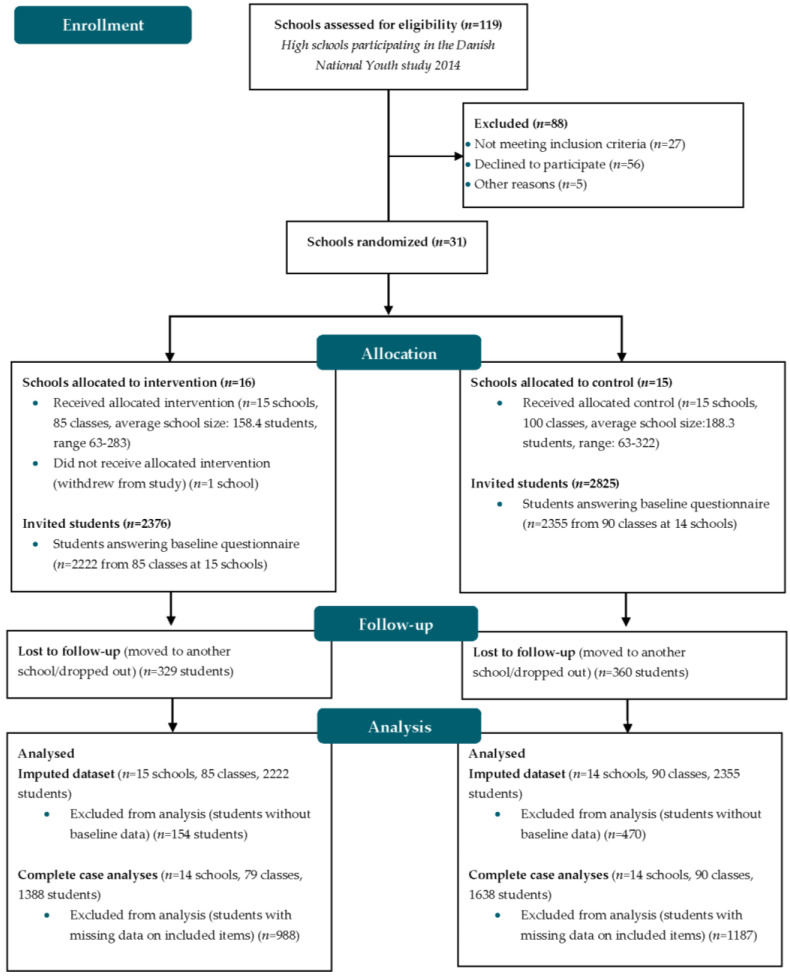
Flow diagram of schools and students in the Healthy High School Study.

**Table 1 ijerph-20-01754-t001:** Overview of the Healthy High School (HHS) intervention.

Intervention Component	Intensity and Duration	Delivered by	Delivered to	Modifiable Determinants of Change
STRUCTURAL POLICIES AND INITIATIVES: 16 initiatives aim to create a healthy and supportive school environment. The initiatives are described in an online catalogue addressed to school management, canteen staff, student councils, teachers, and student counsellors. Four initiatives aim directly at preventing stress:				
1. Development (or revision) of a school stress policy Rationale: To bring focus on stress among students, and to establish guidelines and best practices for stress management at the high school. The high schools were encouraged to adopt a clear action plan that listed tasks, persons responsible for the tasks, and a timeline.	August 2016–June 2017	Student council, school management, teachers, and other relevant school staff	Students (school)	Policies/rules
2. Time management course Rationale: To introduce students to time management tools, e.g., the Eisenhower Decision Matrix, realistic planning, and time tracking. To increase knowledge on how to maintain energy levels throughout the day by, e.g., taking short breaks, being physical active, eating regular meals, and getting enough sleep.	September/October (2 × 45 min)	Student counsellors	Student (class)	Knowledge, awareness, outcome expectations, time management skills
3. Annual course work plans Rationale: To give students the opportunity to manage their time by giving them a complete overview of the annual workload. The coursework plan should include the following information: (1) dates for when the assignments would be started, (2) assignment due dates and time, and (3) expected amount of time to complete the assignments. All high schools were offered free access to an online planning tool.	August 2016–June 2017	High school teachers	Student (class)	School environment, school demands, opportunity, predictability
4. Student counselling Rationale: To support student well-being; identify academic, social, and emotional problems among students, with a specific focus on potential stressors and stress symptoms; address conflicts at home; and to ensure that students received the proper support if needed.	Half-year meetings (September/ October 2016 and February/ March 2017)	Student counsellors	Students (individual)	Formal social support
TEACHING MATERIAL: The teaching material is based on behavior change techniques to change social norms and cognitive factors such as knowledge, awareness, skills, and attitudes. It comprises 17 mandatory lessons (1140 min) for 4 different subjects: Danish, Social Studies, Physical Education, and Sport and Introduction to Natural Science, and optional lessons within a Multi-Subject Coursework (one school week). All lessons are designed to cover official learning goals defined by the Danish Ministry of Education. Activities focusing on stress prevention are included in five lessons:				
Lesson 1: Introduction: Stress, meals, sleep, and physical activity in Introduction to Natural Science. Rationale: To illustrate how stress, meal habits, sleep, and physical activity are interrelated. To raise awareness of own experience of this interrelation.	August 2016 (1 × 45 min)	High school teachers	Students (class)	Knowledge, awareness
Lesson 2: Stress I: Biological stress and a scientific approach to stress measurement methods in Introduction to Natural Science. Rationale: To increase knowledge on the physiology of stress including general stress symptoms and short- and long-term health consequences of stress. To raise awareness of own stress level and stress symptoms. Teachers are to inform about the HHS app.	October 2016 (2 × 45 min)	High school teachers	Students (class)	Knowledge, awareness, outcome expectations
Lesson 3: Stress II: Other aspects and experiments in Introduction to Natural Science. Rationale: See lesson 3 + to introduce students to grounding meditation.	November 2016 (2 × 45 min)	High school teachers	Students (class)	Knowledge, skills, outcome expectations
Lesson 4: Final lesson: Stress, meals, sleep, and physical activity in Introduction to Natural Science. Rationale: Recapitulation and refinement of previous six lessons focusing on the interrelation of stress, sleep, meal habits, and physical activity.	November 2016 (2 × 45 min)	High school teachers	Students (class)	Knowledge
Lesson 5: Stress, individual, and society in Social Studies. Rationale: To increase knowledge of stress, stressors (external and internal stressors), and trends in stress (prevalence of stress among young people). To raise awareness of own stressors.	March 2017 (2 × 45 min)	High school teachers	Students (class)	Knowledge, awareness
PEER-LED INNOVATION WORKSHOP AND DERIVED ACTIVITIES: YOUNG AND ACTIVE: The peer-led innovation workshop aims to inspire high school students to develop and implement new activities to promote physical activity and sense of community at school and did not address stress prevention directly. The workshop was facilitated during school hours by university students in Sport Science and Health. To facilitate and promote establishment of new activities, students could apply for economic support from the research group (up to 40,000 DKK per high school).	August–September 2016 (3 h workshop)	University students in Sports and Health (research assistants)	Students (school)	Knowledge, outcome expectations, attitude, motivation, enjoyment, perceived barriers, self-efficacy, skills (creative and innovation, movement integration), social support, physical environment, financial barriers
THE HHS APP: The HHS app aims to support and promote healthy habits and well-being outside school hours. Stress is addressed through (1) articles about stress, its signs, and symptoms, (2) quizzes, tests, and debunking myths about stress, (3) techniques to prevent and reduce stress, e.g., mindfulness exercises, muscle relaxation exercises, and breathing techniques, and (4) time management techniques (corresponding to the time management course). Moreover, the students could sign up for an eight-week text messaging stress prevention program. One or three times per week, students received a push notification with a tip to manage stress and a suggestion for using relevant functions in the app.	Launched November 2016	Research team or high school staff	Students (individual)	Knowledge, outcome expectations, awareness, attitude, skills

**Table 2 ijerph-20-01754-t002:** Baseline characteristics of students participating in baseline, those lost to follow-up, and imputed cases. Values are percentages followed by numbers in parentheses unless stated otherwise.

Student Characteristics	Students Included in Complete Case Analyses N = 3026		Students Lost to Follow-Up N = 1551		Imputed Cases N = 40 × 4577 ^a^	
	Intervention	Control	Intervention	Control	Intervention	Control
Number of students	1388	1638	834	717	40×2222	40×2355
Gender						
Girls	63.9 (887)	66.4 (1087)	58.1 (484)	59.0 (423)	61.8	64.1
Boys	36.1 (501)	33.6 (551)	41.9 (349)	41.0 (294)	38.3	35.9
Age (years), mean (SD)	16.2 (0.7)	16.2 (1.0)	16.2 (1.1)	16.4 (1.6)	16.2 (0.9)	16.3 (1.2)
Family occupational social class						
High social class (I + II)	50.1 (696)	49.2 (805)	43.1 (359)	42.0 (301)	47.6	47.0
Middle social class (III + IV)	35.9 (498)	33.6 (551)	30.7 (256)	34.6 (248)	34.0	34.0
Low social class (V + VI)	10.0 (139)	11.4 (186)	18.0 (150)	16.6 (119)	12.9	12.9
Unclassifiable	4.0 (55)	5.9 (96)	8.3 (69)	6.8 (49)	5.6	6.2
Immigrant background						
Danish origin	89.8 (1247)	89.4 (1464)	74.3 (620)	80.0 (572)	84.2	86.6
Descendant	8.1 (112)	7.6 (125)	22.0 (183)	14.1 (101)	13.3	9.6
Immigrant	2.1 (29)	3.0 (49)	3.2 (27)	5.6 (40)	2.5	3.8
Major life events within the past year, mean (SD)	1.1 (1.1)	1.1 (1.1)	1.2 (1.2)	1.4 (1.2)	1.1 (1.2)	1.2 (1.1)
Perceived Stress Scale score, mean (SD)	13.4 (6.4)	13.8 (6.3)	14.6 (6.7)	15.2 (6.8)	13.8 (6.5)	14.2 (6.4)
High perceived stress (27–40)	2.8 (39)	3.2 (53)	6.3 (48)	6.2 (40)	4.1	4.0
Moderate perceived stress (14–26)	42.9 (595)	45.5 (745)	45.6 (349)	51.2 (329)	44.2	47.3
Low perceived stress (0–13)	54.3 (754)	51.3 (840)	48.1 (368)	42.6 (274)	51.8	48.7

^a^ Number of students who answered the baseline questionnaire.

**Table 3 ijerph-20-01754-t003:** Effect of the Healthy High School intervention on perceived stress at 9-month follow-up: analyses of the imputed data set and complete cases.

	N	Mean PSS-10 Score (SD) at Follow-Up	Unadjusted Mean Difference between Groups, b [95% CI]	Adjusted ^a^ Mean Difference between Groups, b [95% CI]
Imputed cases	40 × 4577			
Intervention		16.7 (SD = 6.8)	0.1 [−0.74;0.93]	0.42 [−0.16;1.00]
Control		16.7 (SD = 6.7)	0	0
Complete cases	3026			
Intervention		16.5 (SD = 6.9)	0.17 [−0.79;1.13]	0.59 [−0.15;1.33]
Control		16.4 (SD = 6.7)	0	0

^a^ Analyses were adjusted for baseline perceived stress, gender, occupational social class, and major life events.

**Table 4 ijerph-20-01754-t004:** Perceived stress (PSS-10 score) at baseline and follow-up, and effect of the Healthy High School intervention on perceived stress at 9-month follow-up stratified by gender and parental occupational social class.

	Mean PSS-10 Score (SD) at Baseline	Mean PSS-10 Score (SD) at Follow-Up	Imputed Cases Adjusted Mean Difference between Groups, b [95% CI]
Girls			
Intervention	15.3 (6.3)	18.4 (6.6)	0.40 [−0.30;1.10] ^a^
Control	15.4 (6.3)	18.1 (6.5)	0 [reference]
Boys			
Intervention	11.4 (6.1)	14.1 (6.5)	0.37 [−0.32;1.10] ^a^
Control	12.2 (6.1)	14.2 (6.4)	0 (reference)
High Occupational Social Class			
Intervention	13.3 (6.5)	16.3 (6.9)	0.26 [−0.50;1.01] ^b^
Control	14.0 (6.6)	16.6 (6.8)	0 [reference]
Medium Occupational Social Class			
Intervention	14.3 (6.4)	17.2 (6.9)	0.44 [−0.29;1.17] ^b^
Control	14.3 (6.3)	16.8 (6.5)	0 [reference]
Low Occupational Social Class			
Intervention	14.3 (6.8)	17.1 (7.2)	0.32 [−0.92;1.56] ^b^
Control	14.8 (6.1)	17.0 (6.9)	0 [reference]

^a^ Multi-level linear regression adjusted for baseline stress, major life events, and occupational social class; ^b^ multi-level linear regression adjusted for baseline stress, major life events, and gender.

## Data Availability

The datasets generated and/or analyzed during the current study are not publicly available due to the sensitivity of the data but are available from the corresponding author upon reasonable request.

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
