# Peer review of "Preventing Stress among High School Students in Denmark through the Multicomponent Healthy High School Intervention—The Effectiveness at First Follow-Up"

_ijerph, 2023, doi:10.3390/ijerph20031754_

Round 1

Reviewer 1 Report

This article tries to analyze results of a field experiment aimed at reducing stress levels among Danish students. It is also the story of a failure, as the authors admit with admirable intellectual integrity. Thus, in principle I agree with the publication of this article which can be useful in helping the academic community to understand how re-addressing and improving this methodology. Nonetheless, I think that in several parts this article needs extensive clarifications. Namely:

line 92) just add a brief sentence for explaining what DNYS is;

Section 2.2) It is fine that you cite Intervention Mapping protocol without explaining it and sending back the reader to the reference. But, then you cannot write about THE five secondary outcomes taking as given that the reader knows about them. Please, clarify this part. Overall, the section is very confused and overwhelmed and the expected outcomes from the treatments remain somewhat unclear.

Lines 107 and 114) Please, correct the reference Bonnesen at al. unpublished to something the reader could be able to find.

Section 2.6) The way you classified job titles is so unconvincing. You employed a naive arbitrary rule without even providing some examples. Anyway, the way you use this coavariate in the analysis is correct. Nonetheless, you should try to make this categorisation clearer.

Section 3) I really struggled to understand this section, and I am still not sure to have managed it, even if I usually work with the analysis of survey data with multilevel methodology. I am not comfortable both with: i) the table structure, which does not help in figuring out the exact specification of the regression equation; ii) the notation, which sometimes presents absolute values and sometimes differentials, sometimes has standard deviation between brackets, sometimes confidence intervals. Please, find a way to make results description more reader-friendly.

Reviewer 2 Report

Thank you for the opportunity to review the manuscript that addresses a relevant problem in society and among youth. It was difficult to review the manuscript based on the research that entailed a large population and seemingly costly interventions requiring a lot of resources, but yielded no significant impact. The manuscript may surely trigger debate and potential further research based on the barriers that contributed to this lack of apparent changes after the intervention. For me the information is too limited to really reach a conclusion - the research and findings are reported in different manuscripts/ articles referred to and readers might not have time to consult all of these; and the manuscript provides limited information on the actual intervention and its delivery. I would therefore suggest more information on the intervention - perhaps in a table format to indicate the type and brief content of interventions, presenters and time lines. More definite recommendations in the conclusion and recommendations for other researchers may also be of help to enhance the value of the manuscript. It is also not clear if the interventions were based on any previous research evidence, so references to similar interventions can also perhaps be included in the table. This is all I can recommend as a way of enhancing the readability of the manuscript. Not in a position to assess the interventions, I can merely suggest that learners participate by compiling some sort of a personal portfolio, action plan and reflective report that need to be assessed, encouraged and followed up with personal discussions and mentoring where more in-depth interventions are required.  

Round 2

Reviewer 1 Report

The manuscript is sufficiently improved and it can now be take into consideration for publication.